# HDL-Related Parameters and COVID-19 Mortality: The Importance of HDL Function

**DOI:** 10.3390/antiox12112009

**Published:** 2023-11-16

**Authors:** Julia T. Stadler, Hansjörg Habisch, Florian Prüller, Harald Mangge, Thomas Bärnthaler, Julia Kargl, Anja Pammer, Michael Holzer, Sabine Meissl, Alankrita Rani, Tobias Madl, Gunther Marsche

**Affiliations:** 1Division of Pharmacology, Otto Loewi Research Center, Medical University of Graz, Neue Stiftingtalstraße 6, 8010 Graz, Austria; julia.stadler@medunigraz.at (J.T.S.); thomas.baernthaler@medunigraz.at (T.B.); julia.kargl@medunigraz.at (J.K.); anja.pammer@medunigraz.at (A.P.); michael.holzer@medunigraz.at (M.H.); sabine.dirnberger@medunigraz.at (S.M.); alankrita.rani@medunigraz.at (A.R.); 2Gottfried Schatz Research Center for Cell Signaling, Metabolism and Aging, Molecular Biology and Biochemistry, Medical University of Graz, Neue Stiftingtalstraße 6, 8010 Graz, Austria; hansjoerg.habisch@medunigraz.at (H.H.); tobias.madl@medunigraz.at (T.M.); 3Clinical Institute of Medical and Chemical Laboratory Diagnostics, Medical University of Graz, Auenbruggerplatz 15, 8036 Graz, Austria; florian.prueller@uniklinikum.kages.at; 4BioTechMed Graz, 8010 Graz, Austria

**Keywords:** COVID-19, HDL, NMR metabolomics, lipoprotein profiling, cholesterol efflux capacity

## Abstract

COVID-19, caused by the SARS-CoV-2 coronavirus, emerged as a global pandemic in late 2019, resulting in significant global public health challenges. The emerging evidence suggests that diminished high-density lipoprotein (HDL) cholesterol levels are associated with the severity of COVID-19, beyond inflammation and oxidative stress. Here, we used nuclear magnetic resonance spectroscopy to compare the lipoprotein and metabolic profiles of COVID-19-infected patients with non-COVID-19 pneumonia. We compared the control group and the COVID-19 group using inflammatory markers to ensure that the differences in lipoprotein levels were due to COVID-19 infection. Our analyses revealed supramolecular phospholipid composite (SPC), phenylalanine, and HDL-related parameters as key discriminators between COVID-19-positive and non-COVID-19 pneumonia patients. More specifically, the levels of HDL parameters, including apolipoprotein A-I (ApoA-I), ApoA-II, HDL cholesterol, and HDL phospholipids, were significantly different. These findings underscore the potential impact of HDL-related factors in patients with COVID-19. Significantly, among the HDL-related metrics, the cholesterol efflux capacity (CEC) displayed the strongest negative association with COVID-19 mortality. CEC is a measure of how well HDL removes cholesterol from cells, which may affect the way SARS-CoV-2 enters cells. In summary, this study validates previously established markers of COVID-19 infection and further highlights the potential significance of HDL functionality in the context of COVID-19 mortality.

## 1. Introduction

COVID-19, caused by severe acute respiratory syndrome coronavirus 2 (SARS-CoV-2), emerged as a global pandemic in late 2019 [1]. The disease exhibits several clinical manifestations, ranging from mild respiratory symptoms to severe respiratory distress and multi-organ failure [1,2]. Although the primary target of SARS-CoV-2 infection is the respiratory system, the accumulating evidence suggests that COVID-19 is a systemic disease affecting multiple organs and biological processes, and many patients require intensive care unit admission [3,4,5]. Our understanding of the etiology of COVID-19, triggered by the SARS-CoV-2 virus, has seen significant advancement, unveiling intricate molecular pathways and associated metabolic shifts. Notably, COVID-19 is characterized by a profound alteration in metabolism, with distinctive features such as perturbations in the kynurenate/tryptophan pathway and markedly elevated glucose levels [6,7,8]. Furthermore, changes in serum fatty acids [9], carnitines [10], ceramides [11,12] and phospholipids [13,14,15] have been observed. In addition, the metabolism of serum lipoproteins is notably dysregulated, resulting in a pathogenic alteration of both lipoprotein particle size and composition. Low HDL cholesterol and high triglyceride concentrations measured before or during hospitalization are strong predictors of a severe course of the disease [16] and may consequently contribute to an elevated cardiovascular risk [17,18,19,20,21].

Several studies have investigated the relationship between HDL cholesterol levels and COVID-19 outcomes [22,23,24,25,26,27]. Specifically, low levels of plasma HDL cholesterol and alterations in HDL composition have been associated with increased disease severity and worse clinical outcomes in COVID-19 patients [28,29]. Patients with severe COVID-19 symptoms tend to exhibit lower levels of HDL cholesterol than those with milder disease manifestations [24,30,31]. Moreover, individuals with higher antecedent levels of HDL cholesterol have a reduced risk of SARS-CoV-2 infection [27,32]. Our previous findings have established a noteworthy association between the cholesterol efflux capacity (CEC) of HDL and the risk of mortality among patients with COVID-19 [33].

The application of nuclear magnetic resonance (NMR) spectroscopy to COVID-19-positive serum samples has recently revealed significant changes in specific NMR signals, which were identified as glycoprotein moieties (designated as GlycA and GlycB, or Glyc as the sum of both) [34] and supramolecular phospholipid composite signal (SPC) [19]. Both are altered upon COVID-19 infection, but since at least GlyA and GlycB signals primarily arise from acute-phase glycoproteins, among them haptoglobin, α-1-antitrypsin, ceruloplasmin, complement factors C3 and H, and transferrin, they are probably related to various other infectious diseases as well [35]. SPC represents the total NMR signal from the choline head groups of all lipoproteins. The intensity of this signal is proportional to the total amount of SPC in the sample.

By analyzing metabolomics, lipoprotein profiles, GlycA, GlycB, and SPC using NMR spectroscopy, we aimed to gain deeper insights into the complex metabolic changes induced by COVID-19 infection. In our study, we compared the metabolic profiles of COVID-19-infected individuals with individuals with non-COVID-19 pneumonia to identify specific metabolic alterations that are unique to COVID-19 and COVID-19-related mortality.

## 2. Materials and Methods

### 2.1. Study Population and Study Design

We initiated the Alpe_Adria_Coronavirus_Cohort, abbreviated as the ALDOCOV biobank, by collecting residual blood samples from COVID-19 patients whenever these samples were directed to the central laboratory of our university hospital between April and December 2020. After completing all routine laboratory tests, the remaining material was stored at −80 °C until batched analysis. In this retrospective study, we specifically measured the plasma concentrations of interleukin-6 (IL-6), C-reactive protein (CRP), and creatinine. The levels of serum amyloid A (SAA) were quantified using a commercially available kit (Invitrogen, Carlsbad, CA, USA), according to the manufacturer’s instructions.

The database captured essential clinical characteristics, such as antecedent diseases (cardiovascular, oncologic, renal, hypertension, pulmonary, and metabolic conditions like diabetes and obesity), which were recorded for each patient. In addition, anthropometric and clinical data, as well as outcome data, were obtained from the laboratory and hospital information systems. The primary endpoint of the study was death within 90 days after admission, whereas the secondary endpoint was the use of respiratory support with oxygen. This study was conducted with the approval of the institutional ethics committee of the Medical University of Graz (EK 32-475 ex 19/20).

### 2.2. Sample Preparation and NMR Spectroscopy Measurements

Blood serum samples for NMR spectroscopy analysis were prepared as described previously [36]. Briefly, after thawing, 330 µL of the sample was immediately mixed with 330 µL of Bruker’s IVDr NMR buffer for plasma, amongst others containing 3-(trimethylsilyl) propionic acid-2,2,3,3-d4 sodium salt (TSP) (Bruker, Rheinstetten, Germany). Thereof, 600 µL was transferred into 5 mm NMR tubes and measured on the same day. Therefore, tubes were placed into a SampleJet rack (Bruker, Rheinstetten, Germany) holding 96 tubes and placed into the SampleJet (Bruker, Rheinstetten, Germany), where they were stored at 4 °C until further processing. NMR spectra were recorded on a Bruker 600 MHz Avance Neo NMR spectrometer (Bruker, Rheinstetten, Germany). To obtain proton spectra, the measurements were conducted at a constant temperature of 310 K using a standard nuclear Overhauser effect spectroscopy (NOESY) pulse sequence (Bruker: noesygppr1d), a Carr–Purcell–Meiboom–Gill (CPMG) pulse sequence with presaturation during the relaxation delay (Bruker: cpmgpr1d) to achieve water suppression, a standard 2D J-resolved (JRES) pulse sequence (Bruker: jresgpprqf) [37], and a J-edited diffusion and relaxation (JEDI) NMR spectra [38].

### 2.3. Quantification of Analytes Measured by NMR Spectroscopy

Data analysis for lipoprotein quantification was carried out using the Bruker IVDr Lipoprotein Subclass Analysis (B.I.LISA^TM^) method, and small molecular metabolites were quantified using the Bruker IVDr Quantification in Plasma/Serum B.I.Quant-PS^TM^ method. The recently developed PhenoRisk PACS™ RuO method was used to quantify glycoprotein (GlycA, GlycB) and supramolecular phospholipid composite (SPC) signals. This is conducted by sending raw NMR spectrum data to a Bruker online server, which returns 112 lipoprotein parameters (e.g., free and total cholesterol, triglyceride, and total phospholipid content of main classes and subclasses of VLDL, IDL, LDL, and LDL, as well as ApoA-I, ApoA-II, and ApoB protein concentrations), 41 small molecular metabolites (among them amino acids, acids, amines, ketone bodies, glucose, and creatine, as well as potential contaminants/additives like ethanol or EDTA), and GlycA, GlycB, and SPC.

### 2.4. Statistical Analyses

Comparisons between groups were performed using the Mann–Whitney U test. Continuous variables are summarized as medians with interquartile ranges (Q1–Q3), and categorical variables are expressed as absolute frequencies and percentages (%).

To detect alterations in lipoprotein and metabolic profiles, we conducted multivariate statistical analyses, including principal component analysis (PCA) and orthogonal partial least squares discriminant analysis (O-PLS-DA) [39], with associated data consistency checks and 7-fold cross-validation expressed by Q^2^ [40,41] using MetaboAnalyst [42]. Forestplots were created using R.Studio and the forestplot package. All other statistical analyses were performed using GraphPad Prism 8.0 (GraphPad Software, San Diego, CA, USA) and SPSS 26 (SPSS Inc., Chicago, IL, USA). Spearman’s rank-based correlation coefficient was employed for correlation analyses and corrected according to Bonferroni to adjust for multiple testing. Cox regression models were used to find associations between measured parameters and mortality. Information regarding the time of death within 90 days was available for 46 COVID-19 patients. The primary endpoint of death occurred in 28% of the study cohort.

## 3. Results

### 3.1. Baseline Characteristics and Laboratory Results of the Study Cohort

In our study, 47 COVID-19 patients and 31 non-COVID-19 pneumonia patients were included (Table 1). Patient age and the distribution of sex were comparable between the COVID-19 patients and the non-COVID-19 pneumonia control patients. Importantly, the two groups were well matched in terms of inflammatory marker levels of C-reactive protein (CRP), interleukin-6 (IL-6), and serum amyloid A (SAA), thereby ensuring that observed changes were not merely a consequence of varying inflammatory states. Additionally, COVID-19 patients did not exhibit significant differences in levels of the kidney function marker creatinine when compared with the control patients.

Among the COVID-19 patients included in our study, 40% required treatment in the intensive care unit (ICU), reflecting the severity of their condition. Twenty-eight percent of the COVID-19 patients died within 90 days after hospital admission.

### 3.2. Assessment of Metabolites and Lipoproteins in COVID-19

Univariate analyses of the parameters measured by NMR spectroscopy (full list: Appendix A) revealed significant differences between COVID-19 and non-COVID-19 pneumonia patients (Figure 1A). Specifically, phenylalanine, acetoacetic acid, supramolecular phospholipid composite (SPC), HDL cholesterol, HDL-free cholesterol, and HDL-1-free cholesterol (the largest HDL subclass quantified by NMR) were found to be distinct between the two groups (Figure 1B).

Next, we performed multivariate data analyses using orthogonal partial least squares discriminant analyses (OPLS-DA), which allowed clustering for COVID-19 and non-COVID-19 pneumonia patients (Figure 2A) with strong to moderate goodness of fit (correlation coefficient R^2^Y = 0.413 (*p* = 0.003)) and cross-validation score Q^2^ of 0.134 (*p* = 0.008) (Figure 2B). The model reveals some differentiation between groups but not complete separation, which, amongst others, might be due to confounding factors like age, sex, etc. VIP (“variable of importance”) scores of NMR-measured parameters were assessed to obtain the contribution to class separation (Figure 2C). Metabolites with high VIP scores are more important in providing class separation, while those with small VIP scores provide less contribution [43]. The most prominent changes in the measured parameters are shown in Figure 2D, with significant changes in SPC, Glyc/SPC ratio, and phenylalanine. Moreover, HDL-associated phospholipids, total cholesterol, and ApoA-I protein levels are important determinants in discriminating between COVID-19 and non-COVID-19 pneumonia controls.

To provide a comprehensive overview of the differences in all measured lipoprotein parameters of COVID-19 patients compared with non-COVID-19 pneumonia controls, we calculated the respective fold changes and confidence intervals as depicted in Figure 3. The parameters that remain significant after correcting for multiple testing according to the Benjamini–Hochberg method [44] are highlighted in red. Among the main classes, total cholesterol (fold change 0.83, *p* = 0.024), total HDL cholesterol (fold change 0.76, *p* = 0.009), ApoA-I protein (fold change 0.80, *p* = 0.009), and ApoA-II protein (fold change 0.85, *p* = 0.036) exhibited significant differences after correction. However, triglycerides, ApoB-100, and the LDL-to-HDL ratio remained unchanged. Although levels of LDL cholesterol and the ApoB-100 to ApoA-I ratio initially showed significant differences, significance was lost after correction.

Interestingly, we did not observe any differences in the particle number of lipoproteins after correcting for multiple testing (Figure 3B). It must be noted that the online platform we employed for results analysis does not calculate the total particle number of HDL. Triglycerides in the LDL-1 and LDL-6 subclasses initially showed significant differences, but after correction, the significance was lost (Figure 3C). The amount of total cholesterol of HDL, HDL-1 (fold change: 0.71, *p* = 0.032), and HDL-3 subclasses (fold change: 0.78, *p* = 0.019) was significantly lower in COVID-19 patients (Figure 3D). Furthermore, the free cholesterol of HDL (fold change: 0.72, *p* = 0.029) and the HDL-1 subclass (fold change: 0.61, *p* = 0.019) were also significantly lower (Figure 3E). After correction for multiple tests, the phospholipid content of total HDL (fold change: 0.79, *p* = 0.038) was also significantly lower (Figure 3F).

In our analysis, no significant differences were observed in ApoB-100 levels (Figure 3G). However, we did find significant differences in the total ApoA-I (fold change: 0.80, *p* = 0.012), HDL-associated ApoA-I protein (fold change: 0.78, *p* = 0.041), HDL-1 ApoA-I protein (fold change: 0.68, *p* = 0.012), and HDL-3 ApoA-I protein (fold change: 0.86, *p* = 0.016). Additionally, ApoA-II protein levels were significantly different in total HDL (fold change: 0.90, *p* = 0.041) (Figure 3H).

These results highlight the substantial differences in lipoprotein composition in COVID-19-infected patients, particularly lower concentrations of HDL-related parameters.

### 3.3. Correlations between Lipoprotein Parameters and Clinical Data in the COVID-19 Patients

Recent studies have revealed a promising development in the field of COVID-19 prediction. These studies have identified specific signals, currently only detectable by NMR spectroscopy, associated with inflammatory glycoproteins (referred to as Glyc) and their ratio to supramolecular phospholipid composite (known as SPC). Remarkably, these signals are as predictive as established laboratory markers such as ferritin and CRP. This breakthrough has significant potential to improve our ability to predict COVID-19 outcomes [19,20,34,38,45]. Furthermore, we could show that HDL-mediated CEC is inversely linked to mortality risk in COVID-19 patients [33].

We conducted correlation analyses of HDL-related parameters with CEC and inflammatory markers, such as CRP and interleukin-6. Moreover, we included the inflammatory glycoproteins Glyc and SPC in our analyses to study possible associations.

CEC was strongly associated with total ApoA-I protein (r_s_ = 0.693, *p* < 0.001) and HDL-ApoA-I protein (r_s_ = 0.737, *p* < 0.001), but also with HDL cholesterol (r_s_ = 0.559, *p* < 0.001), total ApoA-II protein (r_s_ = 0.583, *p* < 0.001), HDL-free cholesterol (r_s_ = 0.613, *p* < 0.001), and HDL phospholipids (r_s_ = 0.596, *p* < 0.001). Remarkably, there was a strong positive correlation observed between SPC and CEC (r_s_ = 0.661, *p* < 0.001). In contrast, Glyc displayed a notable negative association with CEC (rs = −0.500, *p* < 0.001) (Figure 4).

As anticipated, our results demonstrated a robust positive correlation between CRP levels and inflammatory glycoproteins (r_s_ = 0.722, *p* < 0.001). Conversely, SPC and most HDL-related parameters, except for HDL triglycerides, displayed negative associations. Notably, no significant association was observed between interleukin-6 and the tested variables. Furthermore, creatinine, patient age, and glucose levels were not significantly associated with the selected variables.

### 3.4. Survival Analyses of HDL-Related Parameters in COVID-19

We previously demonstrated a significant inverse association between HDL function, measured as the cholesterol efflux capacity (CEC) of COVID-19 patients on hospital admission, and mortality risk in the same study cohort [33]. In this study, we performed Cox regression analyses on the most prominent COVID-19 discriminatory parameters. These included phenylalanine, acetoacetic acid, the Glyc/SPC ratio, SPC, and lipoprotein parameters, all of which exhibited differences between COVID-19 patients and non-COVID-19 pneumonia controls. The analyses were adjusted for age and sex (and additionally for levels of HDL-C in the case of CEC), and the results are presented as hazard ratios with their corresponding 95% confidence intervals (Figure 5). While we did identify notable correlations between CEC and HDL-related parameters measured by NMR, it is important to note that only CEC exhibited a significant association with mortality risk. Consequently, our findings strongly suggest that CEC stands out as the most promising predictor of mortality when compared to other HDL-related parameters within this particular study cohort.

## 4. Discussion

In this study, we used NMR spectroscopy to assess the metabolomic profile, lipoprotein parameters, and inflammatory markers in COVID-19 patients compared with non-COVID-19 pneumonia patients. Careful matching of the control and COVID-19 groups for inflammatory markers was essential to ensure that differences in lipoprotein levels were specifically attributable to COVID-19 infection, independent of inflammation. Recent research has highlighted disturbances in metabolic pathways associated with COVID-19, particularly in severe cases of the disease. These abnormalities include disturbances in both amino acid and lipid metabolism [15,17,46,47,48]. In addition, the excessive inflammatory response induced by the virus can disrupt the balance of energy homeostasis [49,50,51].

In our initial data analyses employing univariate statistics (with criteria of fold change > 1.33 and FDR-adjusted *p*-value < 0.05), we noted a significant increase in the levels of the amino acid phenylalanine and the ketone body acetoacetic acid in COVID-19 patients when compared to non-COVID-19 pneumonia patients. Previous studies have consistently reported elevated levels of phenylalanine in COVID-19 patients when compared to healthy controls [52,53,54]. Moreover, these increased phenylalanine levels have been associated with disease severity, underlining its potential role as a significant metabolic biomarker for COVID-19 [52,53,54]. It has been suggested that the COVID-19-related increase in pro-inflammatory cytokines triggers muscle breakdown, releasing phenylalanine for gluconeogenesis to meet the increased metabolic demands during infection [52]. The observed increase in the ketone body acetoacetic acid in COVID-19 patients compared with the control group further highlights the presence of an altered metabolic state during infection [17]. In addition to the elevated levels of phenylalanine and acetoacetic acid, our observations revealed a reduction in total HDL free cholesterol, free cholesterol within the largest HDL subclass (HDL-1), and total HDL cholesterol levels in COVID-19 patients compared to non-COVID-19 pneumonia controls. Our data are in good agreement with the results of other studies [16,27,55,56,57].

We developed a classification model employing orthogonal partial least squares discriminant analysis (OPLS-DA) to effectively differentiate between COVID-19 patients and the control group. This model utilized a combination of metabolites, lipoprotein data, and inflammatory parameters as variables. Notably, among the highly discriminative factors, we identified HDL-related parameters such as HDL phospholipids, HDL cholesterol, and ApoA-I. In addition, we observed that SPC levels were significantly lower in the serum of COVID-19 patients than in non-COVID-19 pneumonia patients and were highly indicative of the disease. Consistent with previous research, our results support the notion that SPC and phenylalanine are highly discriminatory biomarkers for COVID-19 [52,53,54]. Moreover, our findings of changes in lipid profiles in COVID-19 patients are in line with previous studies [26,27,58]. Specifically, we observed that total cholesterol, HDL cholesterol, ApoA-I, and LDL cholesterol were all reduced in COVID-19 patients, while the levels of triglycerides were unaffected. After adjusting for multiple testing using the Benjamini–Hochberg correction method, the decrease in LDL cholesterol was no longer statistically significant.

Previous studies reported an increase in VLDL (sub)-fractions in COVID-19 patients [17,18] and an increase in the number of VLDL and IDL particles compared to healthy controls [57,59]. Our findings diverge from these studies, which employed healthy participants as controls. This discrepancy represents a limitation since inflammation itself can exert a substantial influence on plasma lipid levels [17,60,61]. In our study, we did not find significant differences in triglyceride and ApoB levels within lipoprotein subclasses between COVID-19 patients and non-COVID-19 pneumonia controls. Consequently, it is tempting to hypothesize that the impact on VLDL and IDL levels may not be specific to COVID-19 but rather associated with pneumonia or inflammation itself. Nonetheless, it is worth noting that COVID-19 patients exhibited lower levels of ApoA-I, ApoA-II, HDL-1, and HDL-3 cholesterol, which represent large- and medium-sized HDL particles. Consistent with the existing literature, our findings demonstrated a strong correlation between the levels of CRP and inflammatory glycoproteins (Glyc) and the Glyc/SPC ratio [19,62,63,64,65]. When we applied the Bonferroni correction for multiple testing, a robust inverse correlation between CRP levels and HDL-associated parameters emerged. However, no significant correlations were observed between NMR lipoprotein data and creatinine, glucose, or age. This finding highlights the central role of inflammation in shaping both the composition and function of HDL [66,67,68].

HDL particles mobilize plasma membrane cholesterol during cholesterol efflux, which is essential for the proper trafficking and localization of ACE2 receptors, the main entry point for SARS-CoV-2 into cells [69,69]. In a previous study, we evaluated HDL-mediated CEC and found that low HDL-mediated CEC was independently associated with increased mortality risk in COVID-19 patients [33].

It must be noted that macrophages, which play a critical role in cholesterol efflux, probably do not contribute significantly to the cholesterol mass of HDL. For example, transplanting wild-type bone marrow into ABCA1-knockout mice resulted in only a slight increase in plasma HDL cholesterol levels [70]. Because extrahepatic cells require cholesterol but cannot metabolize it, they must efflux it to HDL. The removal of cholesterol from lipid rafts facilitated by HDL could potentially affect the entry of SARS-CoV-2 in several cell types.

Interestingly, the level of ACE2 expression on macrophages varies depending on the subtype and activation state. Importantly, alveolar macrophages have been shown to express higher levels of ACE2 than other macrophage subtypes [71,72,73] and appear to facilitate new viral synthesis.

To elucidate the relationship between CEC and NMR-derived HDL-related parameters, we performed correlation analyses and found the strongest correlation between CEC and HDL-ApoA-I content. Other significantly correlated HDL-related markers included HDL cholesterol, HDL phospholipids, HDL ApoA-II, and SPC.

Surprisingly, in our Cox regression survival analysis, we found no significant association between NMR lipoprotein data and mortality. Therefore, our results highlight the importance of HDL functionality, measured as cholesterol efflux capacity, in relation to COVID-19. Interestingly, the established markers of COVID-19 severity and mortality, such as phenylalanine and the Glyc/SPC ratio [19,45,52], did not show a significant association with mortality in our COVID-19 study cohort, although there was a noticeable trend for phenylalanine.

Our study has several limitations. The main limitation is the small sample size, which may have limited our ability to detect smaller differences in serum metabolites and HDL structure and composition between subjects with or without COVID-19. Furthermore, due to the limited sample size, we were unable to divide the patients into subgroups and further investigate potential confounding factors influencing our results. Another limitation is the observational design, which precludes us from establishing causality.

A key strength of this study is the inclusion of a control cohort of patients with non-COVID-19 pneumonia. This design ensures that differences in metabolites and several HDL-related parameters were specifically attributable to COVID-19 infection, independent of inflammation. In addition, we developed a classification model using orthogonal partial least squares discriminant analysis to discriminate between COVID-19 patients and controls using metabolites and several HDL-related parameters as variables. This model allowed us to gain a deeper understanding of the impact of COVID-19 on HDL metabolism and function.

## 5. Conclusions

Our analysis of lipoprotein parameters revealed the most significant changes in the HDL class, with less pronounced or no changes in other lipoprotein classes. Our findings highlight the importance of assessing HDL functionality rather than just its components. In addition, our study validates established markers of COVID-19 infection versus non-COVID-19 pneumonia, consistently identifying phenylalanine, SPC, and various HDL-related parameters as discriminators. Consequently, the administration of synthetic/recombinant HDL to improve cholesterol efflux capacity (CEC) may be a promising strategy for severely affected COVID-19 patients [74,75,76].

## Figures and Tables

**Figure 1 antioxidants-12-02009-f001:**
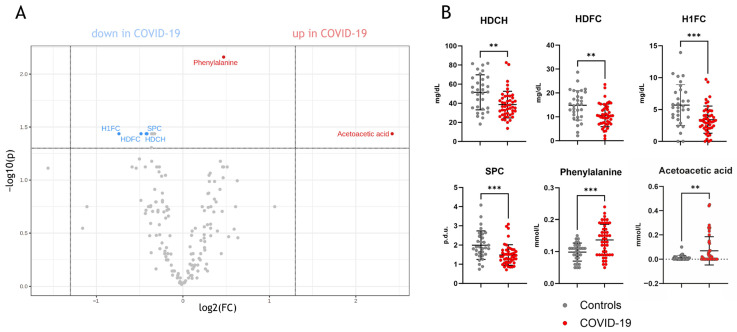
Univariate statistics of NMR-measured parameters in COVID-19 patients compared to non-COVID-19 pneumonia patients. (**A**) shows a volcano plot with significant parameters, which are lower or higher compared to non-COVID-19 pneumonia patients. In (**B**), the individual comparisons of the parameters are shown. HDCH, HDL cholesterol; HDFC, HDL-free cholesterol; H1FC, HDL1-free cholesterol; SPC, supramolecular phospholipid composite; p.d.u., procedure defined units. ** *p* < 0. 01, *** *p* < 0.001.

**Figure 2 antioxidants-12-02009-f002:**
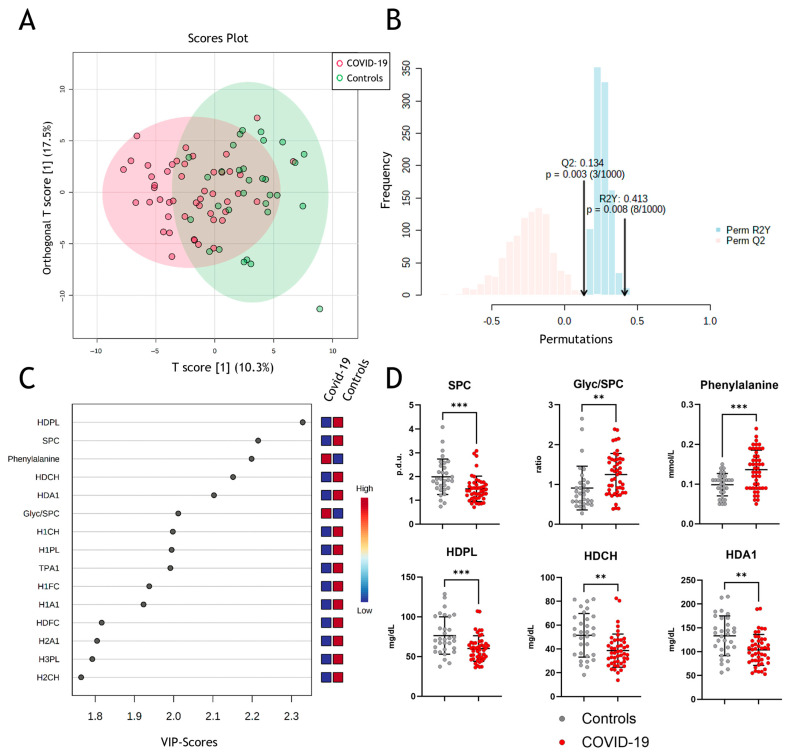
Metabolomic assessment of metabolites in COVID-19. (**A**) Multivariate data analyses of all parameters measured by NMR spectroscopy with orthogonal partial least squares discriminant analyses (OPLS-DA) for differentiation between COVID-19 (red) and non-COVID-19 pneumonia patients (green). (**B**) Permutation validity test: correlation coefficient R^2^Y = 0.413 (*p* = 0.003) and cross-validation score Q^2^ of 0.134 (*p* = 0.008). (**C**) Variable of importance projection scores to obtain the contribution of the parameters to the model. (**D**) Comparison of the most prominent changes between the two groups. SPC, supramolecular phospholipid composite; Glyc, glycoprotein; HDPL, HDL phospholipids; HDCH, HDL cholesterol; HDA1, HDL apolipoprotein A-I; p.d.u., procedure defined units. ** *p* < 0. 01, *** *p* < 0.001.

**Figure 3 antioxidants-12-02009-f003:**
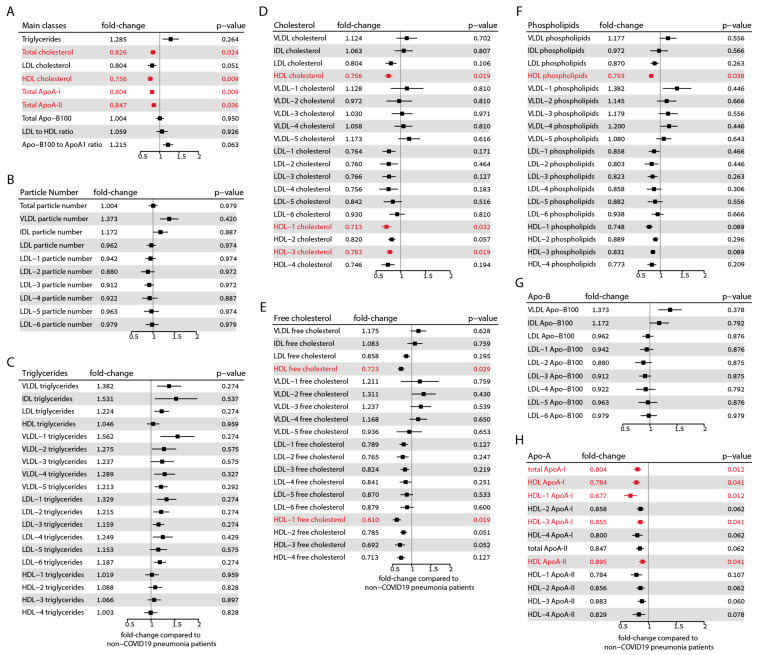
Forest plots showing the alterations observed in lipoprotein parameters between the COVID-19 and non-COVID-19 pneumonia control groups. (**A**) Main classes of lipoproteins. (**B**) Lipoprotein particle numbers. (**C**) Triglyceride distribution in lipoprotein classes. (**D**) Total cholesterol distribution in lipoprotein classes. (**E**) Free cholesterol distribution in lipoprotein classes. (**F**) Phospholipid distribution in lipoprotein classes. (**G**) Apo-B protein distribution in lipoprotein classes. (**H**) ApoA protein distribution in HDL classes. The fold changes to non-COVID-19 pneumonia controls and the corresponding confidence intervals were calculated. Mann–Whitney-U-test-derived *p*-values were corrected according to Benjamini–Hochberg’s procedure. Parameters that remained significant after correction are shown in red.

**Figure 4 antioxidants-12-02009-f004:**
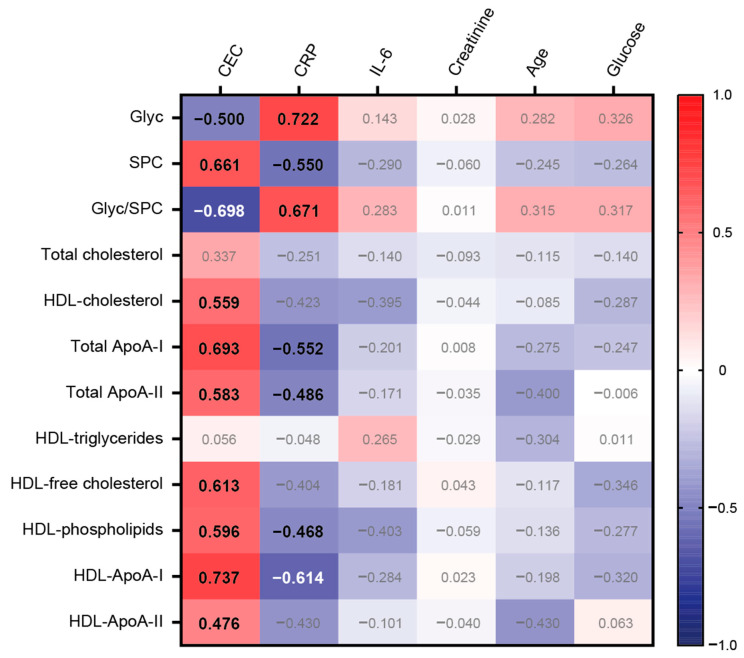
Heatmap representing the correlations between selected variables with CEC and clinical data. Each cell of the heatmap represents a pairwise Spearman correlation between the two parameters indicated in the respective row and column. Correlations that reached significance after the Bonferroni correction are indicated with the corresponding Spearman correlation coefficient in bold. Non-significant correlations are not highlighted.

**Figure 5 antioxidants-12-02009-f005:**
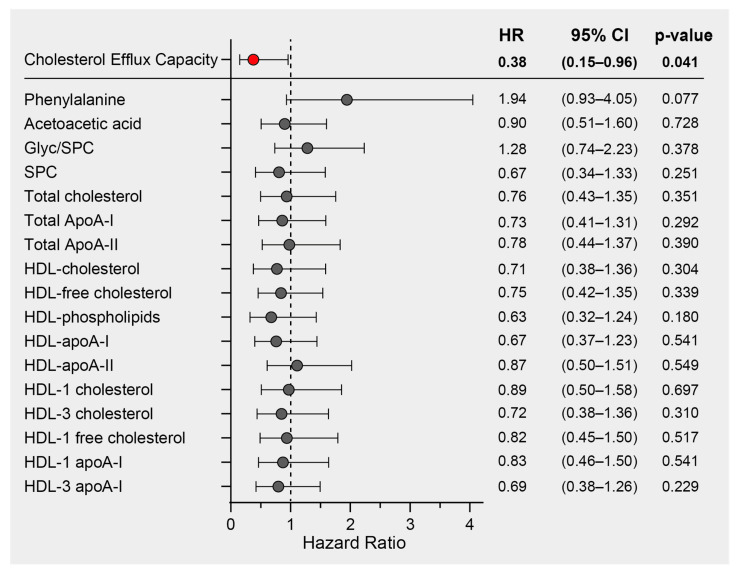
Survival hazard ratios (HRs) per 1 SD increase and 95% confidence intervals (CIs) derived from Cox regression analyses. The association of cholesterol efflux capacity with mortality risk has already been published [33] and was adjusted for age, sex, and HDL-C. The other NMR-derived parameters were adjusted for age and sex.

**Table 1 antioxidants-12-02009-t001:** Baseline characteristics of the study cohort. Values are expressed as median (Q1–Q3). CRP, c-reactive protein; IL-6, interleukin-6; SAA, serum amyloid A; ICU, intensive care unit.

	COVID-19 Patients (*n* = 47)	Non-COVID-19 Pneumonia Patients (*n* = 31)	*p*-Value
Age (years)	68 (56–80)	73 (51–81)	0.716
Female sex	24 (51%)	17 (55%)	0.818
CRP (mg/L)	34.5 (12.2–78.6)	46.3 (4.2–95.2)	0.915
IL-6 (pg/mL)	39.4 (14.5–120)	27.9 (6.7–84.5)	0.309
SAA (µg/mL)	600 (81–3506)	404 (52–1359)	0.124
Creatinine (mg/dL)	0.12 (0.07–0.14)	0.11 (0.09–0.13)	0.927
Hypertension	23 (50%)	14 (45%)	0.203
ICU	19 (40%)	-	
Exitus	13 (28%)	-	

## Data Availability

Data are contained within the article.

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
