# Peer review of "HDL-Related Parameters and COVID-19 Mortality: The Importance of HDL Function"

_antioxidants, 2023, doi:10.3390/antiox12112009_

Round 1

Reviewer 1 Report

Comments and Suggestions for Authors

This study is a follow-on from an earlier study (ref. 33) where the key result was that cholesterol efflux capacity (CEC) was significantly reduced in patients with COVID-19 compared to non-COVID-19 pneumonia, and in particular, CEC was indicative of a fatal course of COVID-19. 

This study is somewhat incremental in its findings, but adds NMR data to reinforce the earlier findings, demonstrating that HDL-related parameters are significantly reduced in patients with COVID-19 pneumonia compared to non-COVID-19 pneumonia. 

I do have one question: Lipoprotein particle number data are measured in Figure 3B, but only total, VLDL, and LDL particles. Why not HDL particle numbers? Am I missing something here?

Author Response

Reviewer 1:

This study is a follow-on from an earlier study (ref. 33) where the key result was that cholesterol efflux capacity (CEC) was significantly reduced in patients with COVID-19 compared to non-COVID-19 pneumonia, and in particular, CEC was indicative of a fatal course of COVID-19. 

This study is somewhat incremental in its findings, but adds NMR data to reinforce the earlier findings, demonstrating that HDL-related parameters are significantly reduced in patients with COVID-19 pneumonia compared to non-COVID-19 pneumonia. 

I do have one question: Lipoprotein particle number data are measured in Figure 3B, but only total, VLDL, and LDL particles. Why not HDL particle numbers? Am I missing something here?

Answer:  We are pleased that our manuscript was favourably received by the reviewer and are happy to consider the helpful comments.

We used the Bruker IVDr Lipoprotein Subclass Analysis B.I.LISA™ online algorithm, which unfortunately does not calculate total particle numbers of HDL. In order to draw attention to this limitation, we have included an additional statement in the results section (Line 212 “Nonetheless, it has to be noted that the online platform we employed for results analysis, does not calculate total particle number of HDL”.)

Reviewer 2 Report

Comments and Suggestions for Authors

Previously published studies have clearly demonstrated that COVID-19 subjects have disturbances in their lipid metabolism and especially reduced HDL concentrations are a typical finding in addition to elevated TG levels. These lipid changes differ depending on the degree of the disease state. The authors of the present study have also previously shown (ref 33) that HDL cholesterol concentrations were lower in COVID-19 subjects when compared to  non-COVID pneumonia patients. In addition in this previous work cholesterol efflux capacity, CEC of HDL of COVID-19 subjects was significantly attenuated. In this manuscript  nuclear magnetic resonance (NMR) spectroscopy was applied to compare the lipoprotein and metabolic profiles of COVID-19-infected patients with non-COVID-19 pneumonia. Importantly, the authors compared the control group and the COVID-19 group using inflammatory markers to be sure that the differences in lipoprotein levels were mostly the consequence of the COVID-19 infection. This is a relevant continuation and a more detailed study using NMR data. The paper is clearly written and contains new data especially on HDL parameters as well as contains a novel discriminating approach, i.e. OPLS-DA method to verify differences between the covid-19 and control subjects. However, there are several issues that need further discussions.

MAJOR COMMENTS

1. The authors showed that the levels of HDL parameters, including ApoA-I, ApoA-II, HDL cholesterol, and HDL phospholipids were significantly reduced in COVID-19 subjects compared to non-COVID-1 pneumonia subjects. The authors use the term supramolecular PL composite, SPC. The term should be explained and in addition why did the authors not use the individual PL species for comparison purposes? There could be significant differences for instance in sphingomyelin or PC levels in HDL particles. This needs further explanations.

2. The authors included here 47 COVID-19 patients and 31 non-COVID-19 pneumonia patients. In their previous study (ref 33) the numbers were 48 and 32. Why deleting one from each group here?

3. The authors give no data whether the COVID-19 and control subjects differ in the following critical conditions: obesity, BMI, metabolic syndrome, pre-diabetes, diabetes, cancer or previous history of CVD since all of these can cause bias in interpreting the data? 

4. It is well known that serum amyloid A, SAA is elevated during inflammation and once binding to HDL particles can kick off apoA-I as well as PON-1. Another important factor is apoC-III that is elevated in inflammation and once attached to HDL can attenuate HDL function in CEC? Any further data on these in addition to that already published (ref 33)?

5. It is somewhat contradictory that there were no differences in VLDL-remnant levels. Reportedly inflammation caused HDL reduction typically causes TRL-remnant elevation. The authors should comment this?

6. CEC is mostly targeted to monocyte-macrophage-foam cell entity, not generally to all plasma membranes. Do macrophage membranes also contain ACE-2 receptors? If so, then removal of membrane free cholesterol via HDL could affect SARS-Cov-2 viral entry process but need careful basic cellular studies to be executed for full proof.

Author Response

Reviewer 2:

Previously published studies have clearly demonstrated that COVID-19 subjects have disturbances in their lipid metabolism and especially reduced HDL concentrations are a typical finding in addition to elevated TG levels. These lipid changes differ depending on the degree of the disease state. The authors of the present study have also previously shown (ref 33) that HDL cholesterol concentrations were lower in COVID-19 subjects when compared to  non-COVID pneumonia patients. In addition in this previous work cholesterol efflux capacity, CEC of HDL of COVID-19 subjects was significantly attenuated. In this manuscript  nuclear magnetic resonance (NMR) spectroscopy was applied to compare the lipoprotein and metabolic profiles of COVID-19-infected patients with non-COVID-19 pneumonia. Importantly, the authors compared the control group and the COVID-19 group using inflammatory markers to be sure that the differences in lipoprotein levels were mostly the consequence of the COVID-19 infection. This is a relevant continuation and a more detailed study using NMR data. The paper is clearly written and contains new data especially on HDL parameters as well as contains a novel discriminating approach, i.e. OPLS-DA method to verify differences between the covid-19 and control subjects. However, there are several issues that need further discussions.

MAJOR COMMENTS

  1. The authors showed that the levels of HDL parameters, including ApoA-I, ApoA-II, HDL cholesterol, and HDL phospholipids were significantly reduced in COVID-19 subjects compared to non-COVID-1 pneumonia subjects. The authors use the term supramolecular PL composite, SPC. The term should be explained and in addition why did the authors not use the individual PL species for comparison purposes? There could be significant differences for instance in sphingomyelin or PC levels in HDL particles. This needs further explanations.

We are pleased that our manuscript was favourably received by the reviewer and are happy to consider the helpful comments.

Answer: We have added some information on the measured parameters, using the Bruker PhenoRisk PACS™ algorithm. (Line 72) “SPC represents the total NMR signal from choline head groups of HDL and LDL subfractions. This signal is more than the sum of the relevant phospholipids and is an independent signal that can be detected by NMR.”
Unfortunately, different classes of lipoprotein-associated phospholipids cannot be measured by the NMR spectroscopy system used in our study.

  1. The authors included here 47 COVID-19 patients and 31 non-COVID-19 pneumonia patients. In their previous study (ref 33) the numbers were 48 and 32. Why deleting one from each group here?

Answer: We had to exclude two serum samples from the analyses due to insufficient serum volume for NMR measurements.

  1. The authors give no data whether the COVID-19 and control subjects differ in the following critical conditions: obesity, BMI, metabolic syndrome, pre-diabetes, diabetes, cancer or previous history of CVD since all of these can cause bias in interpreting the data? 

Answer: We appreciate the reviewer's comment. Due to the low number of patients included in our study, the statistical power to analyze the potential influence of additional confounding factors was not possible. We have added a sentence to the limitations section acknowledging this limitation (line 404).

  1. It is well known that serum amyloid A, SAA is elevated during inflammation and once binding to HDL particles can kick off apoA-I as well as PON-1. Another important factor is apoC-III that is elevated in inflammation and once attached to HDL can attenuate HDL function in CEC? Any further data on these in addition to that already published (ref 33)?

Answer: As correctly noted by the reviewer, HDL structure, composition, but also post-translational modifications (oxidation, carbamylation) may all affect cholesterol efflux capacity of HDL. To study HDL proteomic and lipidomic composition in detail, HDL must be carefully isolated. Due to the small sample volume available, we were unable to isolate HDL for further analysis and therefore we have no additional data on apoC-III levels in this study cohort.

  1. It is somewhat contradictory that there were no differences in VLDL-remnant levels. Reportedly inflammation caused HDL reduction typically causes TRL-remnant elevation. The authors should comment this?

Answer: As correctly noted by the reviewer, HDL also obtains lipids, especially PLs, from other lipoproteins. When triglyceride-rich lipoproteins are hydrolyzed, their surface PLs (and apolipoproteins) are shed and acquired by HDL. During inflammation, lipoprotein lipase activity is inhibited, increasing VLDL-remnant levels. Our cohorts were matched for inflammation markers and also VLDL-remnant levels were similar in both groups. However, the decrease in HDL-related parameters, especially in the COVID-19 group, suggests an additional COVID-specific effect. This may be due to the fact that COVID-19 can also damage the liver, and the liver is important for producing and maintaining HDL levels. Our observation is in line with several other studies that have shown that COVID-19 patients have lower HDL levels than healthy controls. Additionally, some studies have shown that liver function tests are abnormal in a significant proportion of COVID-19 patients.

  1. CEC is mostly targeted to monocyte-macrophage-foam cell entity, not generally to all plasma membranes. Do macrophage membranes also contain ACE-2 receptors? If so, then removal of membrane free cholesterol via HDL could affect SARS-Cov-2 viral entry process but need careful basic cellular studies to be executed for full proof.

 Answer: We appreciate the reviewer's comment. It has to be noted that macrophages, which play a critical role in cholesterol efflux, probably do not contribute significantly to the cholesterol mass of HDL. For example, transplanting wild-type bone marrow into ABCA1-knockout mice resulted in only a slight increase in plasma HDL cholesterol levels (doi: 10.1172/JCI12810). Because extra-hepatic cells require cholesterol but cannot metabolize it, they must efflux it to HDL. The removal of cholesterol from lipid rafts facilitated by HDL could potentially affect the entry of SARS-CoV-2 in several cell types. Interestingly, the level of ACE2 expression on macrophages varies depending on the subtype and activation state. Importantly, alveolar macrophages have been shown to express higher levels of ACE2 than other macrophage subtypes (DOI: 10.1126/signal.abq1366, doi: 10.2147/JIR.S300747, https://www.nature.com/articles/s41598-022-07918-6) and appear to facilitate new viral synthesis.

We have added this to the discussion (line 379)

Round 2

Reviewer 2 Report

Comments and Suggestions for Authors

The authors cover well all of my queries and I do not have further comments.